# Safety Assessment of *Bacteroides Uniformis* CECT 7771, a Symbiont of the Gut Microbiota in Infants

**DOI:** 10.3390/nu12020551

**Published:** 2020-02-20

**Authors:** Eva M. Gómez del Pulgar, Alfonso Benítez-Páez, Yolanda Sanz

**Affiliations:** Microbial Ecology, Nutrition & Health Research Unit, Institute of Agrochemistry and Food Technology, Spanish National Research Council (IATA-CSIC), 46980 Valencia, Spain

**Keywords:** safety, probiotics, *Bacteroides uniformis*, inflammation, liver, obesity, metabolic function

## Abstract

The formulation of next-generation probiotics requires competent preclinical studies to show their efficacy and safety status. This study aims to confirm the safety of the prolonged oral use of *Bacteroides uniformis* CECT 7771, a strain that protected against metabolic disorders and obesity in preclinical trials, in a sub-chronic 90 day trial in animals. The safety assessment was conducted in male and female Wistar rats (*n* = 50) administered increasing doses (10^8^ CFU/day, 10^9^ CFU/day, or 10^10^ CFU/day) of *B. uniformis* CECT 7771, 10^10^ CFU/day of *B. longum* ATCC 15707^T^, which complies with the qualifying presumption of safety (QPS) status of the EU, or vehicle (placebo), as the control. Pancreatic, liver, and kidney functions and cytokine concentrations were analyzed. Bacterial translocation to peripheral tissues was evaluated, and colon integrity was investigated histologically. No adverse metabolic or tissue integrity alterations were associated with treatments; however, alanine aminotransferase levels and the ratio of anti-inflammatory to pro-inflammatory cytokines in serum indicated a potentially beneficial role of *B. uniformis* CECT 7771 at specific doses. Additionally, the microbial community structure was modified by the interventions, and potentially beneficial gut bacteria were increased. The results indicated that the oral consumption of *B. uniformis* CECT 7771 during a sub-chronic 90 day study in rats did not raise safety concerns.

## 1. Introduction

Knowledge of the gut microbiota structure and function has dramatically increased in recent years through the use of next-generation sequencing technologies (NGS) [1]. These techniques have also helped to unravel how interactions among the gut microbiota, the environment (e.g., diet, habitat, stressors, etc.), and the host (e.g., age, genetics, etc.) may impact health-related outcomes. This information has led to the identification of microbiome-based biomarkers of disease risk and possible health-promoting bacteria (usually known as probiotics) with future clinical and nutritional applications.

Until now, the probiotic bacteria used to maintain health status and prevent or alleviate diseases in humans have consisted mainly of *Lactobacillus* spp. and *Bifidobacterium* spp. [2]. Strains of these species isolated from fermented foods and human or animal biological samples now have a long history of safe use in foods and qualify for qualifying presumption of safety (QPS) status [3]. Nonetheless, the use of NGS techniques has allowed the identification of other intestinal bacteria in humans, including abundant endogenous microbiota that could play differential biological roles in both gastrointestinal and extraintestinal pathologies [4,5,6]. Although those unveiled species and strains are predominant and recurrent inhabitants of the human gut, unlike classical probiotic species [7,8], they lack a history of use for human consumption. Therefore, a specific safety assessment must be completed, guaranteeing their unequivocal security status according to their intended purpose [9,10].

The Bacteroidetes taxon represents between 20 and 40% of the colonic bacteria and is considered one of the most abundant phyla of the human gut microbiota. This phylum includes the *Bacteroides* genus, which consists of 5–18% of the total microbiota, a percentage higher than that represented by the well-known genera *Lactobacillus* and *Bifidobacterium* in the adult gut [11,12]. It can be expected that members of this genus play an essential role in the gut environment and impact human physiology. Nevertheless, research on the potential function of members of this genus to act as probiotics is only in the initial stages. Some species of the genus *Bacteroides* have already shown the ability to outcompete pathogens [13], attenuate auto-immunity by optimizing Th1/Th2 balance and by inducing T reg cell differentiation [14], and positively influence satiety and glucose metabolism [5,15]. These effects have been mediated, at least in part, by the host-microbe interactions relying on bacterial structural components, such as polysaccharide A (PSA) of *Bacteroides fragilis* [16], or bacterial metabolites, such as short-chain fatty acids (SCFAs: acetate, butyrate, and propionate) [17]. SCFAs are mainly derived from the fermentation of complex carbohydrates carried out directly by *Bacteroides* sp. or via cross-feeding mechanisms between *Bacteroides* and other intestinal bacteria, such as *Faecalibacterium prausnitzii*, *Roseburia* spp., and *Anaerostipes* spp. [18].

*Bacteroides uniformis* CECT 7771 was initially isolated from the feces of healthy breastfed infants [19], who have been demonstrated to have a higher abundance of this *Bacteroides* species than formula-fed infants. This strain was selected among other *Bacteroides* species and strains due to its ability to induce an anti-inflammatory cytokine profile in macrophage cultures in vitro and to ameliorate the metabolic dysfunction of diet-induced obesity in mice. The oral administration of *B. uniformis* CECT 7771 reduced body weight gain and improved lipid metabolism, diminishing liver steatosis and serum cholesterol and triglyceride levels in obese mice. This strain also reduced leptin levels and improved glucose metabolism since it decreased fasting concentrations of serum glucose and insulin, therefore improving glucose tolerance [5].

Previously, we performed a preliminary evaluation of the safety and tolerance to the oral administration of *B. uniformis* CECT 7771 in a short-term acute study (six days) in healthy and immunocompromised mice, and the strain showed no adverse effects at a dosage of 10^9^ CFU/day [20].

This study aimed to further evaluate the tolerance to the oral administration of *B. uniformis* CECT 7771 in a sub-chronic study (lasting 90 days in total) in rats by conducting an exhaustive examination of possible adverse effects, including assessments of body weight and fat, food intake, bacterial translocation, intestinal tissue histology, and biochemical and immune markers.

## 2. Methods

### 2.1. Bacterial Strain and Culture Conditions

The *B. uniformis* CECT 7771 strain was recovered, isolated, and characterized from the stools of healthy infants upon informed consent to their parents [19] and deposited in the Spanish Type Culture Collection (CECT). The bacteria were grown in liquid Schaedler medium (without hemin) at 37 °C in an anaerobic chamber (Bactron 300-2, Shellab, Cornelius, OR, USA). Cells were harvested by centrifugation (6000× *g* for 10 min) and washed with phosphate-buffered saline solution (PBS, 130 mM sodium chloride, 10 mM sodium phosphate, pH 7.4). Bacteria were then resuspended in 10% sterile skim milk for animal trials (Scharlau, Barcelona, Spain). Aliquots of these suspensions were immediately frozen in liquid nitrogen and stored at −80 °C until use. After freezing and thawing, the number of live cells was determined by counting colony-forming units (CFUs) on Schaedler agar medium plates after 48 h of incubation at 37 °C in anaerobiosis. Cell viability was 98.3%. One fresh aliquot was thawed daily to avoid differences in culture viability during the study.

### 2.2. Sub-Chronic (90-Day) Oral Toxicity Study in Rats

The sub-chronic toxicity study was designed based on the specifications of the OECD Guideline to perform a repeating dose, 90 day oral toxicity study (408) [21] in 9-week-old male and female Wistar rats (Envigo RMS, S.L. Sant Feliu de Codines, Barcelona). During the adaptation period (7 days), animals of the same sex were housed in cages in groups of 2–3 animals per cage to create 5 groups (*n* = 10:5 males and 5 females per group). The animals were held in a temperature-controlled (23 °C) room with a 12 h light/dark cycle with 40–50% relative humidity. Standard procedures were carried out to minimize the inter-individual microbiota variation in the animal studies [22]. To this end, soiled cage bedding was mixed daily and exchanged among all cages for 1 week prior to treatment initiation. This procedure favored the establishment of uniform gut microbiota in the rats as a baseline, as previously reported [22,23].

Animals were fed a conventional laboratory diet (Purified Rodent Diet AIN-76A) and water ad libitum. The animals were divided into different experimental groups that also received one of the following supplements daily: (1) a control group receiving placebo (10% (w/v) skimmed milk) (control); (2) a group receiving 1 × 10^10^ CFU/day *B. longum* ATCC 15707^T^ (B_longum10), a strain with a QPS status [10]; (3) a group receiving 1 × 10^8^ CFU/day *B. uniformis* CECT 7771 (B_Unif8); (4) a group receiving 1 × 10^9^ CFU/day *B. uniformis* CECT 7771 (B_Unif9); and (5) a group receiving 1 × 10^10^ CFU/day *B. uniformis* CECT 7771 (B_Unif10). The bacterial doses were administered after incorporation into a piece of sweet jelly (Transgel®, Charles River, Wilmington, MA, USA), thus avoiding the use of gavage feeding and reducing the stress in animals. Signs of toxicity, morbidity, and mortality were recorded twice daily, and individual body weights were recorded once per week. Food consumption was assessed weekly by registering food weight at the beginning and the end of each week.

After the 90 day intervention, the animals were anaesthetized, and blood was collected from each rat by aortic puncture, which was then immediately killed by cervical dislocation. The large and small intestine, liver, and mesenteric lymph nodes (MLNs) were removed and stored for different analyses as described below. All procedures involving animals were approved explicitly by the Ethics Committee of the University of Valencia (Animal Production Section, Central Service of Support to Research—SCSIE, University of Valencia, Spain) and authorized by “Dirección General de Agricultura, Ganadería y Pesca Generalitat Valenciana” (Authorization Number: 2017/VSC/PEA/00125). A study design scheme is illustrated in Figure 1.

### 2.3. Bacterial Translocation

Bacterial translocation was analyzed in the blood (serum) and homogenized MLN. Samples were diluted in buffered peptone water (1 g/mL). A total of 100 μL of the resulting sample was plated to count bacterial growth using Schaedler agar media (Oxoid, Basingstoke, U.K.) for *Bacteroides* and Wilkins-Chalgren anaerobe agar media (Oxoid, Basingstoke, U.K.) for total anaerobe quantification after incubation at 37 °C in an anaerobic chamber for three days. In the case of bacterial translocation (growth on agar plates), the bacterial colonies were recovered, and the bacterial DNA was extracted using a MasterPure Gram-positive DNA Purification Kit (Epicentre-Illumina, San Diego, CA, USA). Then, the possible presence of *B. uniformis* CECT 7771 in the plates was assessed by PCR using specific oligonucleotides for the glutamate decarboxylase gene of this specific species [17]. PCRs contained Phusion High-Fidelity Taq Polymerase (Thermo Scientific, Wilmington, USA), 0.5 µM forward oligonucleotide (BU-544F: 5’-TATGCAACCAAGCTGATGAACGAAG-3’), 0.25 µM reverse oligonucleotide (BU-544R-r: 5’-AGAGGTTGGCCACGATGTTGATAC-3’), 0.25 mM dNTPs, and 20 ng of DNA sample. The gene was amplified with 25 PCR cycles consisting of the following steps: 95 °C for 20 s, 63 °C for 30 s, and 72 °C for 60 s. PCR products were visualized with 1% agarose gel electrophoresis.

### 2.4. Determination of Cytokine Concentrations

The cytokines IL-10, IFN-γ, and TNF-α were quantified in serum and jejunum samples by using Simplex Luminex kits for each immune parameter and ProcartaPlex Basic Rat kits (eBioscience, Vienna, Austria) in a Luminex 100 IS™ (Luminex Corporation. Austin, TX, USA). The jejunum samples (100 mg) were first manually disaggregated with sterile scalpels, and the resulting tissue sections were homogenized in 0.5 mL RIPA buffer containing a cocktail of protease inhibitors (Complete, Mini tablets, Roche Life Science, Mannheim, Germany). Homogenization was completed by using a TissueRuptor® device (Qiagen, Hilden, Germany), and the samples were kept on ice during the procedure. Then, the samples were centrifuged (16,000× *g* for 10 min at 4 °C), and the protein concentration of the resulting supernatant was quantified and adjusted to 10 mg/mL for cytokine assessment as recommended by the manufacturer. These measurements were carried out in duplicate for each sample.

### 2.5. Biochemical Parameter Analysis

Biochemical parameters were evaluated in serum samples by blood centrifugation (1000× *g* for 10 min at room temperature). We assessed the following parameters by using the respective enzymatic assay kits: alkaline phosphatase (ALP), alanine aminotransferase (ALT), amylase (BioVision Incorporated, Milpitas, USA), urea, and creatinine (Sigma-Aldrich (Merck KGaA), Darmstadt, Germany). These measurements were carried out in duplicate for each sample and following the manufacturers’ instructions.

### 2.6. Histology and Histometry

Sections of the colon from each animal were collected immediately after sacrifice and were rapidly fixed in 4% paraformaldehyde in phosphate-buffered saline (PBS) at pH 7.4 for 24 h at 4 °C. After that, the colon sections were dehydrated in a graded series of ethanol, cleared with xylene, and embedded in paraffin to obtain serial microtome sections (3 μm thick). Then, the samples were stained with hematoxylin/eosin (HE) solutions to evaluate the structural aspects of the colon (crypts, villi, colonocytes, epithelium, etc.). For histometry evaluations, the number of goblet cells per intestinal villi and the depth of intestinal crypts of Lieberkhün (10 fields measured per sample) in the HE-stained sections were determined. In addition, the lymphoid area of individual follicles containing Peyer’s patches was examined in the HE-stained colon sections as a determinant of the status of the gut-associated lymphoid tissue (GALT). Measurements were obtained from images obtained using a light microscope fitted with a Nikon, Olympus, Eclipse 90i, U.K., camera using NIS Elements BR 2.3 research software (Kingston, Surrey, U.K.).

### 2.7. Statistical Analysis

All data were analyzed using GraphPad Prism 5.01 (Graph Pad Software Inc., San Diego, CA, USA). The results are expressed as the mean ± SEM. All datasets were first analyzed to check whether data were normally distributed by using the Kolmogorov–Smirnov normality test and Bartlett’s test to assess the homogeneity of variance. For parametric datasets, two-way analysis of variance (ANOVA) was applied to compare means, and Tukey’s multiple comparison test was used as a post hoc test for pairwise comparisons. For non-parametric data, a Kruskal–Wallis test was performed to compare medians, and Dunn’s multiple comparison test was used as a post hoc test for pairwise comparisons. Differences between groups were established using an unpaired Student’s *t*-test (when the data were normally distributed) or the Mann–Whitney U-test (when the data were non-normally distributed). Differences were established when *p* ≤ 0.050 in all analyses.

### 2.8. Fecal Microbiota

Fecal samples were collected at the end of the intervention period. Feces (spontaneously evacuated) from a total of forty-two rat specimens was collected before the euthanasia procedure. Approximately 200 mg stools were used for DNA extraction using a Power Fecal DNA Isolation Kit (Qiagen, Hilden, Germany) following the manufacturer’s instructions. A diluted aliquot of the fecal DNA was prepared at ~20 ng/μL to be used in PCRs. Approximately 20 ng of DNA (1 μL of diluted DNA) were used to amplify the V3-V4 hypervariable regions from the bacterial 16S rRNA gene by a 25 cycle PCR program consisting of the following steps: 95 °C for 20 s, 55 °C for 20 s, and 72 °C for 20 s. The PCR was performed using Phusion High-Fidelity Taq Polymerase enzyme (Thermo Scientific) and 6-mer barcoded primers, which target a wide range of bacterial 16S rRNA genes: S-D-Bact-0341-b-S-17 (CCTACGGGNGGCWGCAG) and S-D-Bact-0785-a-A-21 (GACTACHVGGGTATCTAATCC). Dual-barcoded amplicons that consisted of approximately 500 bp fragments were purified from triplicate reactions per sample using the Illustra GFX PCR DNA and Gel Band Purification Kit (GE Healthcare, Little Chalfont, U.K.). Amplicon DNA was quantified using a Qubit 3.0 fluorometer and the Qubit dsDNA HS Assay Kit (Life Technologies, Carlsbad, CA, USA). Samples were mixed by combining equimolar quantities of amplicon DNA (100 ng of each sample) and sequenced in an Illumina MiSeq platform with a 2 × 300 PE configuration (CNAG, Barcelona, Spain).

### 2.9. Next-Generation Sequencing Data Analysis

Raw data were delivered in fastq paired-end files. Paired-end assembly and sample de-multiplexing were carried out using sequence information from the respective 6-mer DNA barcodes and PCR primer sequences as a mapfile input in the qiime suite of analysis (v1.9.1) [24]. In a similar manner, chimaera detection, alpha diversity, beta diversity, and operational taxonomic unit (OTU)-picking approaches were conducted using respective Python scripts implemented in qiime v1.9.1. Non-parametric Kruskal–Wallis and Mann–Whitney U-tests were applied to compare respective variable distributions among groups. Treatment and sex were used as categorical variables in beta diversity analysis based on the Bray–Curtis and Jaccard dissimilarity indexes. Differences among groups were established when the log2-fold-change was larger than 1.5 and the *p*-value was less than 0.10 after adjusting by multiple testing correction methods, including FDR (false discovery rate).

## 3. Results

### 3.1. General Health, Food Intake, and Weight of the Animals

During the experimental procedure, there were no noticeable changes in regular activity, hair luster, or behavior in any of the groups of rats. Animals did not suffer from diarrhea or other sickness related to treatment. At the end of the study, all animals were alive and healthy in all experimental groups. There were no meaningful differences in body weight gain or loss or feed intake among the different groups (Appendix A) or when males and females were analyzed separately (Appendix A). We detected no compelling differences in the weight of livers among the groups (Appendix A), not even when males and females were studied separately (Appendix A). The coloration, size, and appearance of the analyzed tissues were normal, and no differences were observed among the experimental groups.

### 3.2. Bacterial Translocation

Bacterial growth was observed upon incubation at 37 °C under anaerobic conditions in some MLN homogenate samples and blood from both the experimental and placebo groups. Following a species-specific PCR assay, no colonies were identified as *B. uniformis* CECT 7771, since the fragment corresponding to the glutamate decarboxylase gene was not detected when compared to a positive control using a sample from a pure culture of the strain. This finding indicated that despite the relatively high bacterial concentration administered to rats, the bacterium was not able to translocate from the intestinal lumen to the MLN or bloodstream of rats.

### 3.3. Biochemical Parameters

Potential deleterious effects on different organs due to the oral administration of *B. uniformis* CECT 7771 were evaluated with biochemical parameters related to pancreatic, liver, and kidney functions. ALT levels were significantly reduced in the B_Unif9 and B_Unif10 groups compared to the control group (*p* < 0.050 and *p* < 0.010, respectively) (Figure 2 A). This pattern may indicate a dose-dependent effect of *B. uniformis* CECT 7771 on ALT levels. When the serum values of the ALT of males and females were analyzed separately, we observed that the highest dose of *B. uniformis* CECT 7771 significantly reduced the levels of the enzyme compared to the placebo control and *B. longum* ATCC 15707^T^ (*p* < 0.050 in both comparisons) in males (Figure 2 C). A similar trend was observed for females, but the difference was not equally robust (Figure 2B). ALP (Appendix A), creatinine (Appendix A), and urea concentrations (Appendix A) were not different between the controls and experimental groups fed *B. uniformis* CECT 7771, even when the males and females of each group were considered independently (Appendix A).

Amylase activity was similar in all the groups (Figure 3A). However, when males and females were considered independently, we found that females had meaningful differences. Amylase activity was decreased in the B_Unif9 group compared to the *B. longum* group (*p* < 0.050), the B_Unif8 group (*p* < 0.010), and B_Unif10 group (*p* < 0.010), but the differences were not consistent with the existence of a dose-response relationship (Figure 3B). No differences were observed among the groups for males (Figure 3C).

### 3.4. Cytokine Concentrations in Serum and Jejunum

The concentration of the anti-inflammatory cytokine IL-10 in serum was significantly reduced in the B_longum10, B_Unif8, and B_Unif9 groups compared to the control group (*p* < 0.050, *p* < 0.010, and *p* < 0.010, respectively), while there were no significant differences between the control group and the B_Unif10 group (*p* > 0.100) (Figure 4A). When males and females were analyzed independently, we noted that the IL-10 concentration was significantly increased in the B_Unif10 group compared to the B_Unif9, B_Unif8, and B_longum10 groups (*p* < 0.050 for all comparisons) in the serum of females (Appendix A). The IL-10 concentration of serum samples collected from males was increased in the B_Unif10 group (*p* < 0.050) compared to the B_longum10, B_Unif8, and B_Unif9 groups (Appendix A). The IL-10 concentration of jejunum samples showed similar levels as in serum when all animals were considered together (Figure 4B), but when males and females were considered separately, some differences could be identified (Appendix A).

The IFN-γ concentrations in serum samples were lower in the B_longum10, B_Unif8, and B_Unif9 groups (*p* < 0.050 in all cases) than in the control group (Figure 4C). Nonetheless, when serum samples from only females were considered, no differences in serum IFN-γ concentrations were detected (Appendix A). Consequently, samples from the males in the B_longum10 and B_Unif9 groups showed lower IFN-γ levels (*p* < 0.050) than those from males in the control group (Appendix A). The IFN-γ concentration in jejunum samples was lower in the B_longum10, B_Unif8, and B_Unif9 groups than in the control group (*p* < 0.050 for all comparisons). Additionally, a significant reduction in jejunum IFN-γ was detected in the B_Unif9 group compared to the B_Unif8 group (*p* < 0.001) (Figure 4D). When the male and female data were analyzed separately, a reduction in the cytokine concentration of jejunum samples from females was found in all the study groups compared to the control group (*p* < 0.050 for all comparisons), except in the B_Unif10 group (Appendix A), whereas no differences were observed in jejunum samples from males (Appendix A).

As the representative anti-inflammatory cytokine IL-10 and the pro-inflammatory cytokine IFN-γ regulate one another, the ratio of IL-10/IFN-γ was estimated to obtain information about the balance between anti- and pro-inflammatory responses to the treatment [25,26,27]. We observed significantly higher ratios in the serum of the B_longum10 and B_unif9 groups than in the serum of the B_Unif8 group (*p* < 0.010) (Figure 5A). Male and female IL-10/IFN-γ serum ratios were analyzed separately, and similar trends were observed for the B_Unif9 group in females (Appendix A) and the B_longum10 reference group in males, but the differences did not support marked alterations in cytokine balance (Appendix A). The IL-10/IFN-γ ratios in jejunum samples had similar patterns to those in the serum samples; the ratios were higher in the B_Unif9 and B_longum10 groups compared to the control group (*p* < 0.001 for both comparisons). A similar trend was detected for the B_longum10 reference group compared to the B_Unif8 group (*p* < 0.001) (Figure 5B). In females, the jejunum IL-10/IFN-γ ratio was increased in the B_Unif9 group compared to the control group (*p* < 0.010) (Appendix A), whereas in males, only the jejunum IL-10/IFN-γ ratio was significantly higher in the reference group B_longum10 compared to the control group (*p* < 0.050) (Appendix A). Extremely low TNF-α concentrations (below limit of detection (LOD)) were detected in serum and jejunum samples from rats, possibly due to technical limitations. Therefore, we could not reliably determine any change in this inflammatory marker among the groups.

### 3.5. Colon Mucosal Histology

The effect of the intervention on colon tissue structures is shown in Appendix A. There were no significant differences in the parameters measured among the experimental groups when considering males and females together or separately.

Table 1 summarizes the main results of the body-, organ function-, and inflammation-associated variables analyzed in the treatment groups and control group.

### 3.6. Effect of the Intervention on Fecal Microbiota Composition

The *B. uniformis* CECT 7771 intervention in rats did not result in major changes in the number of bacterial species in the fecal microbiota. Accordingly, differences in the five common ecological descriptors used to assess alpha diversity, including the observed richness, Chao’s richness, Shannon’s diversity, Simpson’s evenness, and Simpson’s reciprocal index, were not detected when all groups were compared. However, the beta diversity analysis revealed significant changes in the structure of the microbial community as a result of the administration of the *B. uniformis* CECT 7771 strain. The Permanova test on distances obtained from the Bray–Curtis dissimilarity index (pseudo-F = 1.197, *p*-value = 0.054) and Jaccard index (pseudo-F = 1.099, *p*-value = 0.005) indicated that the different treatments induced different shifts in the gut microbial community of rats. A distance-based redundancy analysis (dbRDA) indicated that rats fed *B. uniformis* CECT 7771 exhibited almost a consensus change in the gut microbiota, as these samples were clustered towards the negative values of the constrained ordination dimension CAP2 (Appendix A). This analysis also showed that the effects on the microbiota were rather different when rats were fed *B. longum* or placebo. At the taxonomy level, we also found that microbial species belonging to the Christensenellaceae family were significantly increased in the B_longum10, B_unif9, and B_unif10 treatment groups compared to the control group (*p* < 0.050). Additionally, members of the Bacillaceae family were found to be more abundant in the B_longum10 and B_unif10 groups (*p* < 0.050). At the genus level, we found that *Christensenella* species were more abundant in the B_longum10, B_unif9, and B_unif10 groups than in the control group (*p* = 0.017, *p* = 0.05, and *p* = 0.033, respectively). The proportion of reads assigned to *Bacillus* species was observed to be significantly higher in the B_longum10 and B_unif10 groups than in the control group (*p* = 0.014).

At the OTU level, we detected consistent results with the taxonomy assessment at the family and genus levels. For instance, OTUs related to the *Bifidobacterium* genus were more abundant in the B_longum10 group than in the control group, showing more than a four log2-fold-change (OTU4426298, log2-fold change = 4.65, *p* < 0.012; OTU670, log2-fold change = 4.14, *p* < 0.012). The analysis of OTUs at the genus and species level also revealed that *Christensenella*-associated OTUs were more abundant in the B_longum10 (OTU552235 and OTU940, log2-fold change = 2.08 and 1.93, and *p* = 0.027 and 0.046, respectively) and B_Unif10 groups (OTU552235, log2-fold change = 1.74, *p* = 0.083) than in the control group. An increase in *Akkermansia muciniphila*-associated OTUs was also detected in the B_Unif10 group (OTU593043, log2-fold change = 2.78, *p* = 0.083) with a concomitant increase in *Lactobacillus*- (OTU581474, log2-fold change = 1.8, *p* = 0.092) and *Bifidobacterium*-associated OTUs (OTU4426298 and OTU670, log2-fold change = 5.14 and 4.79, and *p* = 0.001 and 0.001, respectively). These differential features were also partially replicated in the B_Unif8 (*Lactobacillus* OTUs) and B_Unif9 (*Bifidobacterium* OTUs) groups. OTUs closely related to *B. uniformis* species presented an increase only in the B_Unif10 group (OTU181953 and OTU572476, log2-fold change = 1.70 and 1.73, respectively, *p* = 0.097).

## 4. Discussion

Currently, commercialized probiotics are primarily composed of *Lactobacillus* and *Bifidobacterium* strains isolated from human or animal biological samples or derived from fermented foods, and these strains are widely consumed in different countries and cultures due to their generally recognized safety status [28,29]. The identification of new components of the intestinal microbiota, due to next-generation DNA sequencing technologies and advances in culturing methodologies, has increased the possibility of evaluating the functional role of new endogenous intestinal bacteria that can constitute so-called “next-generation probiotics” [30]. Some of these new potential probiotics under comprehensive evaluation belong to the *Bacteroides* genus [31,32]. Notably, some of these *Bacteroides* species are related to a lean phenotype in human observational studies [15,33], although contradictory results have also been published [34]. Additionally, the bacteria belonging to this genus may show differential biological effects depending on the species and strain [31,32].

In particular, we selected a strain of the species *B. uniformis* because breastfeeding has been demonstrated to both reduce the risk for developing obesity and type-2 diabetes [35] and increase the abundance of *B. uniformis* in the fecal microbiota of healthy infants compared to formula feeding [1,19]. Likewise, *B. uniformis* CECT 7771 exhibited higher anti-inflammatory potential in vitro than other human infant *Bacteroides* species and strains. The effectiveness of this strain has been evaluated successfully in an animal model of high-fat diet-induced obesity [5]. The oral administration of *B. uniformis* CECT 7771 improved metabolic dysfunction in obese mice. In addition, preliminary evaluation of the safety of this strain was performed after acute oral administration to mice for six days, showing no adverse effects on animals [20]. We completed toxicological and safety assessments of *B. uniformis* CECT oral administration by performing a 90 day study in Wistar rats, given the potential use of this bacterial strain for alleviating obesity complications in humans.

The oral sub-chronic toxicity evaluation showed that rats treated with high dosages (up to 1 × 10^10^ CFU/day) of the potentially probiotic strain *B. uniformis* CECT 7771 were healthy after daily oral administration for 90 days. A dosage of 1 × 10^10^ CFU/day in rats is equivalent to 1.62 × 10^9^ CFU/day in humans when normalized to body surface area. Throughout the study, no adverse effects were observed in body weight, food intake, or other general indicators of the animals’ health status (behavior, luster, etc.) as a consequence of *B. uniformis* CECT 7771 oral administration.

One of the issues that should be addressed in safety assessments is the absence of bacterial translocation from the intestinal lumen to MLN and blood since this may constitute a risk for systemic infection (bacteremia). Therefore, the possibility of the translocation of the strain evaluated was investigated as an indication of its potential pathogenicity [36]. In the previous evaluation of *B. uniformis* CECT 7771’s safety in an acute toxicity study, no translocation events were associated with the treatment [20]. Likewise, no translocation of *B. uniformis* CECT 7771 to MLN or blood was detected in the *Bacteroides*-treated groups in the present study. Although some bacterial counts were found in MLN, these were detected in both the control and experimental groups, suggesting that this finding might be due to cross-contamination as described in previous studies [37]. This notwithstanding, we could not disregard the idea that bacterial species recovered from the MLN could be the result of microbe trafficking mediated by dendritic cells from the intestine to the MLNs via the lymphatic system [38,39]. In any case, specific PCRs for the detection of the *Bacteroides uniformis* CECT 7771 glutamate decarboxylase gene in the isolated colonies were all negative.

Biochemical parameters were assessed in serum to detect potential adverse sub-clinical effects of *B. uniformis* CECT 7771 on the function of different organs. Kidney function based on the quantification of urea and creatinine indicated no detrimental effects of *B. uniformis* CECT 7771 administration. The ALT and ALP activities, indicators of liver function [40,41], showed no drastic alterations in the activity of the liver, and the results obtained, particularly for ALT activity, suggested a dose-dependent protective effect of *B. uniformis* CECT 7771 on liver function regardless of rat sex [41]. However, further studies in animal models of human liver diseases and using additional markers of hepatic function are necessary to confirm this hypothesis. The activity of serum amylase has been shown to be lower in female rats than in male rats [42], and we observed that a dose of 1 × 10^9^ CFU of *B. uniformis* CECT 7771 decreased the levels of the enzyme exclusively in females. This result suggested a sex- and dose-specific effect of the strain. The oral administration of classical probiotics of the genera *Bifidobacterium* and *Lactobacillus* (~10^9^ CFU) also decreased amylase activity in male rats [43,44]. This notwithstanding, the lack of information on the effects stratified by sex from previous animal studies made it impossible to draw more specific conclusions from previous probiotic studies. In fact, this variable should be considered in future animal studies since these specific microbe-host interactions have already deserved special attention in human studies.

As expected, cytokine concentrations were generally low (IL-10 and IFN-γ), and TNF-α was below the LOD in all experimental groups. This result could indicate that an inflammatory state was not induced by bacterial administration; however, we should not disregard technical limitations such as the utilization of a no high-sensitivity detection kit or issues during sample preparation procedures according to previous reports [45,46]. The IL-10/IFN-γ ratio was used as an indicator of the balance between the anti- and pro-inflammatory responses to the treatment. The results suggested that *B. uniformis* at 10^9^ CFU/day and *B. longum* at 10^10^ CFU/day induced an anti-inflammatory cytokine profile, especially in the intestine. Similar response trends could be identified in males and females when considered separately, although there were subtle differences, most likely due to the modest sample size of the subgroups rather than the different sexes.

The analysis of the composition of rat fecal microbiota showed no changes in richness and diversity between groups. Reductions in such parameters have often been associated with different disorders, such as inflammatory bowel diseases [47], obesity, and obesity co-morbidities in other studies [48]. This notwithstanding, we found changes in the global structure of the microbiota among groups according to beta diversity analyses. These differences were probably due to the differential abundances detected in particular taxonomy clades among groups. For example, in the B_longum10 group, we observed an increase in *Bifidobacterium* species, which indicated that this bacterial strain could transiently colonize or reach the large intestine. A similar effect was observed in the B_Unif9 and B_Unif10 groups, where a significant increase in *B. uniformis*-associated OTUs was detected as a result of the treatments. This observation suggested that these doses of *B. uniformis* CECT 7771 may be necessary for the successful transient colonization of the rat intestine (10^9^ and 10^10^ CFU/day) since differences were not detected at a lower concentration. However, future studies should be conducted to demonstrate that viable *B. uniformis* CECT 7771 cells can be recovered from rat feces after similar treatments, since in this study, only molecular methods were applied. Interestingly, *Christensenella* species and *Akkermansia muciniphila* were increased in the B_Unif10 group compared with the control group. Both species have been proven to exert beneficial effects in preclinical studies related to obesity or associated with healthy metabolic phenotypes in human observational studies [49,50]. Our results suggested a potential interaction among these species that could also contribute to some of the beneficial effects attributed to *B. uniformis* CECT 7771 in disease models [5]. Therefore, future studies should be conducted to demonstrate the metabolic cross-talk among these microbial species.

In conclusion, the results of the safety evaluation of the sub-chronic oral administration of *B. uniformis* CECT 7771 to rats for 90 days did not raise safety concerns. No adverse effects were recorded regarding general health indicators, gut mucosal histology, bacterial translocation, organ function, or immune response. The analysis of fecal microbiota indicated that *B. uniformis* CECT 7771, as well as the bacterium used as the QPS control (*B. longum* ATCC 15707^T^) could reach the large intestine and modulate the gut microbiota composition without adverse effects on richness and diversity and without promoting the invasion of recognized pathogens. The results also indicated improvements in markers related to mucosal inflammatory milieu inflammation (anti-/pro-inflammatory cytokine ratio) and liver function as a result of *B. uniformis* CECT 7771 administration. 

## 5. Conclusions

We conclude that the oral consumption of *B. uniformis* CECT 7771 during a sub-chronic 90 day study in rats did not raise any safety concerns. According to the lack of safety concerns and the potential benefits identified, tentatively, a dosage of 10^9^ CFU/day further extrapolated to a human equivalent dose could be recommended for future clinical trials. In this regard, additional studies are required to confirm the safety and efficacy of this bacterial strain in humans and to prove its role in obesity and related metabolic disorders definitively.

## Figures and Tables

**Figure 1 nutrients-12-00551-f001:**
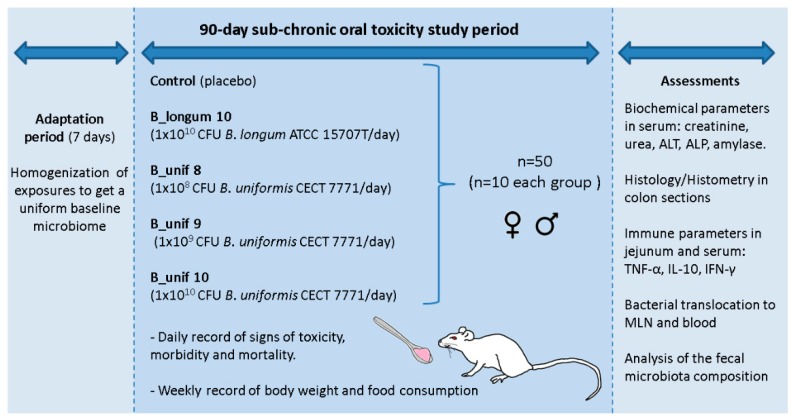
Schematic representation of the study protocol to evaluate the safety of *Bacteroides uniformis* CECT 7771 in a sub-chronic (90-day) oral toxicity study in rats. B_Unif, *B. uniformis*; MLN, mesenteric lymph node. ALT: alanine aminotransferase, ALP: alkaline phosphatase, TNF-α: tumor necrosis factor-alpha, IL-10: interleukine-10, IFN-γ: interferon-gamma, ♀: rat females, ♂: rat males.

**Figure 2 nutrients-12-00551-f002:**
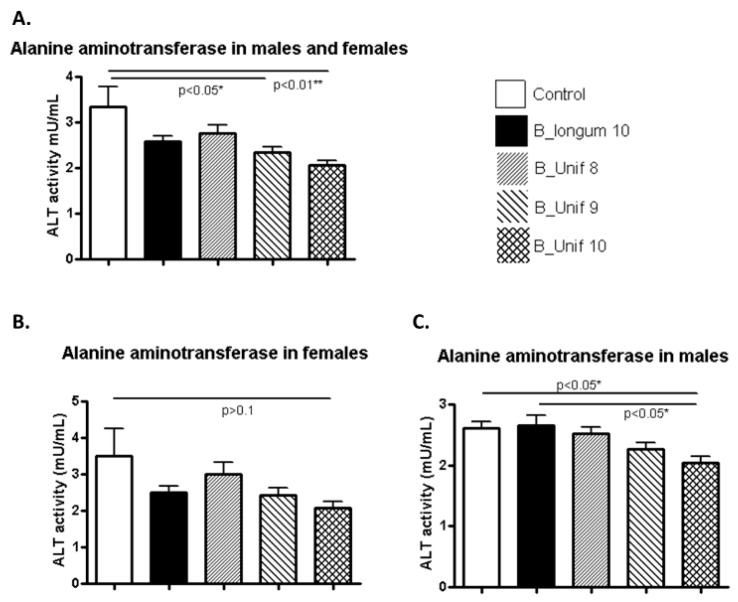
Alanine aminotransferase (ALT) activity measured in the serum of the experimental groups. (**A**) ALT activity (mU/mL) at the end of the study for all groups (50 animals, *n* = 10 per group). (**B**) ALT activity (mU/mL) at the end of the study for females in all groups (25 animals, *n* = 5 per group). (**C**) ALT activity (mU/mL) at the end of the study for males in all groups (25 animals, *n* = 5 per group). Measurements were made in duplicate. Statistical significance was considered when *p* < 0.050. * *p* ≤ 0.05, ** *p* ≤ 0.01.

**Figure 3 nutrients-12-00551-f003:**
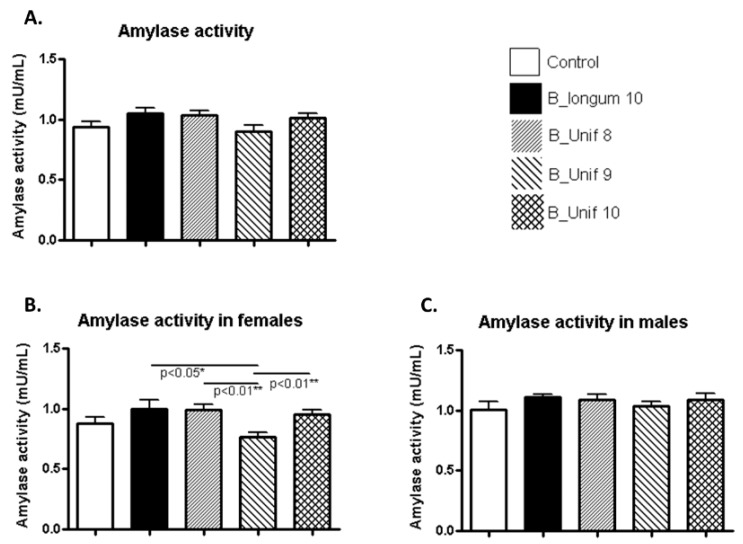
Amylase activity measured in the serum of the experimental groups. (**A**) Amylase activity (mU/mL) at the end of the study for all groups (50 animals, *n* = 10 per group). (**B**) Amylase activity (mU/mL) at the end of the study for females in all groups (25 animals, *n* = 5 per group). (**C**) Amylase activity (mU/mL) at the end of the study for males in all groups (25 animals, *n* = 5 per group). Measurements were made in duplicate. Statistical significance was considered when *p* < 0.050. * *p* ≤ 0.05, ** *p* ≤ 0.01.

**Figure 4 nutrients-12-00551-f004:**
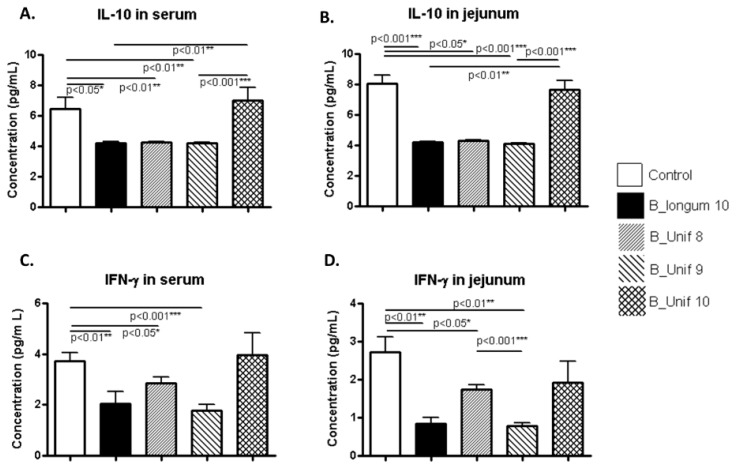
IL-10 and IFN-γ concentrations in serum and jejunum samples from the experimental groups. (**A**) IL-10 serum concentration (pg/mL) at the end of the study for all groups (50 animals, *n* = 10 per group). (**B**) IL-10 jejunum concentration (pg/mL or pg in 10 mg of tissue) at the end of the study for all groups (50 animals, *n* = 10 per group). (**C**) IFN-γ serum concentration (pg/mL) at the end of the study for all groups (50 animals, *n* = 10 per group). (**D**) IFN-γ jejunum concentration (pg/mL or pg in 10 mg of tissue) at the end of the study for all groups (50 animals, *n* = 10 per group). Measurements were made in duplicate. Statistical significance was considered when *p* < 0.050. * *p* ≤ 0.05, ** *p* ≤ 0.01.

**Figure 5 nutrients-12-00551-f005:**
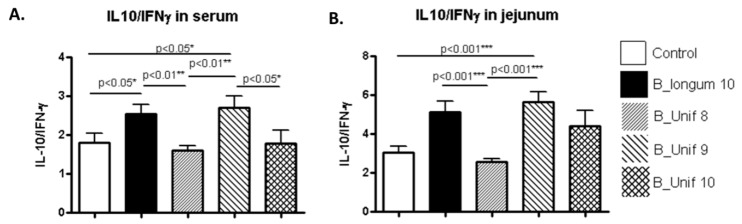
IL-10/IFN- γ ratios in serum and jejunum samples from the experimental animal groups. (**A**) IL-10/IFN-γ ratios (pg/mL) in serum samples at the end of the study for all groups (50 animals, *n* = 10 per group). (**B**) IL-10/IFN-γ ratios (pg/mL) in jejunum samples at the end of the study for all groups (50 animals, *n* = 10 per group). Measurements were made in duplicate. Statistical significance was considered when *p* < 0.050. * *p* ≤ 0.05, ** *p* ≤ 0.01.

**Table 1 nutrients-12-00551-t001:** Summary of different assessments to support the safety of *B. uniformis* CECT 7771.

	B_longum10			B_unif8			B_unif9			B_unif10		
	♀	♂	Both	♀	♂	Both	♀	♂	Both	♀	♂	Both
Physiology												
Weight gain	ns	ns	ns	ns	ns	ns	ns	ns	ns	ns	ns	ns
Liver weight	ns	ns	ns	ns	ns	ns	ns	ns	ns	ns	ns	ns
Food intake	ns	ns	ns	ns	ns	ns	ns	ns	ns	ns	ns	ns
Liver function												
Serum ALT	ns	ns	ns	ns	ns	ns	ns	ns	YES↓	ns	YES↓	YES↓
Serum ALP	ns	ns	ns	ns	ns	ns	ns	ns	ns	ns	ns	ns
Kidney function												
Serum creatinine	ns	ns	ns	ns	ns	ns	ns	ns	ns	ns	ns	ns
Serum urea	ns	ns	ns	ns	ns	ns	ns	ns	ns	ns	ns	ns
Pancreas function												
Serum amylase	ns	ns	ns	ns	ns	ns	ns	ns	ns	ns	ns	ns
Cytokine production												
Serum IL-10	ns	YES↓	YES↓	ns	YES↓	YES↓	ns	YES↓	YES↓	ns	ns	ns
Jejunum IL-10	ns	YES↓	YES↓	ns	YES↓	YES↓	ns	YES↓	YES↓	ns	ns	ns
Serum IFN-γ	ns	YES↓	YES↓	ns	ns	YES↓	ns	YES↓	YES↓	ns	ns	ns
Jejunum IFN-γ	YES↓	ns	YES↓	YES↓	ns	YES↓	YES↓	ns	YES↓	ns	ns	ns
Serum IL-10/IFNγ	ns	YES↓	YES↓	ns	ns	ns	ns	YES↓	YES↓	ns	ns	ns
Jejunum IL-10/IFNγ	ns	YES↓	YES↓	ns	ns	ns	YES↓	ns	YES↓	ns	ns	ns
Colon mucosa architecture												
Lieberkhün crypt depth	ns	ns	ns	ns	ns	ns	ns	ns	ns	ns	ns	ns
Goblet cells/crypt	ns	ns	ns	ns	ns	ns	ns	ns	ns	ns	ns	ns

The table compiles the overall results of the variables analyzed compared to control rats. Comparisons of each sex (females (♀) and males (♂)) and in combination (both) are shown. ns: no pairwise difference found between groups. YES: difference in value distributions. The arrow indicates the direction of change, being lower (↓) or higher (↑).

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
