# Peer review of "Safety Assessment of Bacteroides Uniformis CECT 7771, a Symbiont of the Gut Microbiota in Infants"

_nutrients, 2020, doi:10.3390/nu12020551_

Round 1

Reviewer 1 Report

A very interesting work with a lot of data presented.

Some minor remarks:

In Figure 3; amylase activity in females: What is the most relevant explanation for the significant differences observed in the females and not in the males group?

Again, what is the meaning of the significant reduction in the alanine transferase values observed in both males and females group?

Based on the results of the study what could be the preferred dose of the probiotic used in this experimental work and why? 

Author Response

Reviewer#1 concerns:

1) In Figure 3; amylase activity in females: What is the most relevant explanation for the significant differences observed in the females and not in the males group?

NOTE: Please be aware that line numbers should differ between the original MS-formatted version of the document submitted and the journal-formatted version.

R/ According to our literature review, there are reference intervals established for several biochemical blood variables in rats. Houtmeyers and coworkers described important sex-associated differences in at least ten different variables explored, one of those the amylase concentration in blood that was reported to be lower in females than in males (Houtmeyers et al. 2016. doi: 10.1111/vcp.12419). Therefore, we hypothesised that the lower value of amylase detected in females could also be influenced in a dose-specific manner by B. uniformis CECT 7771. Probiotic strains of the species Lactobacillus also exerted decreases in amylase activity in rats after controlled administration to males (doi: 10.1016/j.anaerobe.2013.08.006, doi: 10.1186/1476-511X-10-116) previously. Unfortunately, the lack of information on whether or not differences on amylase activity could be attribute to the sex of the animals make it impossible to do comparisons with previous publications in this regard. Notwithstanding, we think the results of our study when discriminating by sex, will pave the way for future studies taking into account this variable that is already a matter of investigation in human interventions. We have added additional information in the new version (lines 410-418).

2) Again, what is the meaning of the significant reduction in the alanine transferase values observed in both males and females group?

R/ ALT (alanine aminotransferase) is a classical biochemical marker for the rapid evaluation of liver function. The enzyme responsible for such function is highly concentrated in liver cells with almost null levels in the blood. Therefore, when the liver is damaged, ALT is released into the blood and the level increases permitting an early detection of liver dysfunction. Consequently, any meaningful change in comparison with the control group would indicate possible detrimental effects of treatments on liver physiology (if ALT increases), thus rising potential safety concerns over the oral administration of B. uniformis CECT 7771. We observed no elevation of ALT levels in any treatment. By contrast, we did detect decreasing levels in a dose-dependent manner in males and females rats. These results would indicate that oral administration of B. uniformis CECT 7771 would confer protection against liver damage regardless of the rat sex. The ideas above were already disclosed in lines 405-410, but we have included slight changes in this part of the text to better clarify this issue.

3) Based on the results of the study what could be the preferred dose of the probiotic used in this experimental work and why?

R/ Considering the risk/benefit balance (the lack of safety concerns by oral administration of the B. uniformis CECT 7771 and the potential beneficial effects observed), we think that the adequate dose of this bacteria for future studies should be between 109 and 1010 CFU/day. We have included a short statement indicating so in lines 461-465.

Reviewer 2 Report

This manuscript describes some positive effects exerted by supplementation to adult rats with different dosages of Bacteroides uniformis CECT 7771, a strain isolated from infant gut microbiota, previously used successfully by the Authors in a trial of obese mice. Overall the paper is well written and conceived, however some concerns and inaccuracies are listed below. In general, I notice that there are too many data not shown and many Supplementary Figures. I would suggest to show all the significant differences in Figures/Tables in the paper, and not as Supplementary material.

Major Revisions

Abstract

-Line 13 (and line 74 of Introduction). Explain the meaning of “sub-chronic”.

-line 17. I suggest to replace “immune response” with “cytokine secretion”, considering that no other parameters have been studied.

Methods

2.2. Sub-chronic (90-day) oral toxicity study in rats

-Lines 96-98. Unclear sentence. What does it mean “rotation between uncleaned cages”?

2.3. Bacterial translocation

-Line 133. Here it is stated that the primers are strain-specific, whereas in Results, line 236, it is reported “species-specific”. Please, correct as appropriate.

2.7. Statistical analysis

-Lines 172-173. Other than normality, was also homogeneity of variance checked by appropriate statistical tests, before performing the ANOVA?

2.8. Fecal microbiota

-Line 181. Why only 42 instead of 50?

Figure 1 and 3.4 paragraph title, in Results

In these two parts it is reported that cytokines were analysed in ileum, whereas elsewhere in Results and in Figures 4 and 5 it is written “jejunum”. Please, correct as appropriate.

Figures

-Figures 2-6. Remove titles in the Figures, not necessary: panel letters and legends below Figures are self-explanatory. Uniform Figure legends: in Figures 2-5 there are symbols with rectangles, whereas in Figure 6 all the informations, i.e. name of bacteria and concentrations, are given below the plots. Uniform “Ref” and “B_longum 10”. If the Authors choose the symbol rectangles, make them bigger, as it is hard to distinguish the 3 different B. uniformis concentrations.

-Figure 4 B-D. Concerning cytokine concentration in jejunum, what does it mean “pg/mL”? mL of what? Should instead the data be normalised to µg tissue, or proteins?

-Figure 5. How exactly has been this ratio calculated? Should it be the ratio between the two cytokine concentrations? If so, it should be a dimensionless number, with no measure unit (pg/mL). Please, explain.

Results

3.2. Bacterial translocation

-Lines 239-241 and lines 396-398 of Discussion. The Authors should justify the cross-contamination. They cite in the Discussion only one reference as another example of cross-contamination, but it is just one study and a very very old one (year 1990).

3.4. Cytokine concentrations in serum and illeum

-Line 278 and Figure 4A. Are the Authors sure that the difference between Control and B. uniformis 8 is not significant?

-Line 306. Truncated sentence. Please, complete.

-Line 320. Is this result in accordance with other literature data?

3.6. Effect of the intervention on the faecal microbiota composition

It seems a little bit strange that, except for Christensenellaceae and Bacillaceae families, the majority of data described in this paragraph are not shown in Figures/Tables. Why? In particular:

-Lines 328-321. No data shown on alpha diversity? Same comment applies to lines 344-358.

Minor Revisions

Introduction

-Line 43. Delete the first “specific”, as it is repeated twice.

-Line 59. Change “with” to “and”.

-Line 71. Put superscript “109

Discussion

-Line 362. Change “its” to “their”.

Author Response

Reviewer#2 concerns:
This manuscript describes some positive effects exerted by supplementation to adult rats with different dosages of Bacteroides uniformis CECT 7771, a strain isolated from infant gut microbiota, previously used successfully by the Authors in a trial of obese mice. Overall the paper is well written and conceived, however some concerns and inaccuracies are listed below. In general, I notice that there are too many data not shown and many Supplementary Figures. I would suggest to show all the significant differences in Figures/Tables in the paper, and not as Supplementary material.

NOTE: Please be aware that line numbers should differ between the original MS-formatted version of the document submitted and the journal-formatted version.

R/. We have incorporated Table 1 that presents briefly the primary results of most assessments performed. Notwithstanding, we will maintain all supplementary material as they are extensively cited in the main text and in case the readers want to review these results in detail.
Major Revisions
Abstract

-Line 13 (and line 74 of Introduction). Explain the meaning of “sub-chronic”.

R/ In medicine, a chronic disease is described as a condition lasting more than three months. Therefore, our animal trial of oral administration during 90 days will be better described as a sub-chronic study. We have added new information for clarifying this aspect in the revised version (line 82).

-line 17. I suggest to replace “immune response” with “cytokine secretion”, considering that no other parameters have been studied.

R/ Done as suggested (line 22-23).

Methods
2.2. Sub-chronic (90-day) oral toxicity study in rats
-Lines 96-98. Unclear sentence. What does it mean “rotation between uncleaned cages”?

R/ We have re-written this paragraph to improve the understanding of procedures performed to minimise the inter-individual microbiota variation in animal studies (see lines 106-110).

2.3. Bacterial translocation
-Line 133. Here it is stated that the primers are strain-specific, whereas in Results, line 236, it is reported “species-specific”. Please, correct as appropriate.

R/ The primers are “species-specific” given this gene is also present in other B. uniformis strains (e.g. ATCC 8492) according to BLAST searches (line 141). This error has been corrected.

2.7. Statistical analysis
-Lines 172-173. Other than normality, was also homogeneity of variance checked by appropriate statistical tests, before performing the ANOVA?

R/ The results from different assessments done in GraphPad software did include the analysis of homogeneity of variance. Consequently, we have included a brief statement mentioned it (lines 184-185).

2.8. Fecal microbiota
-Line 181. Why only 42 instead of 50?

R/ As we mentioned in the text, only 42 fecal samples (naturally evacuated) could be recovered from respective rats the morning previously to euthanasia procedure (line 194-95).

Figure 1 and 3.4 paragraph title, in Results
In these two parts it is reported that cytokines were analysed in ileum, whereas elsewhere in Results and in Figures 4 and 5 it is written “jejunum”. Please, correct as appropriate.

R/ Cytokines quantification was performed in jejunum samples. Figure 1 and subheading (line 265) were changed accordingly.

Figures
-Figures 2-6. Remove titles in the Figures, not necessary: panel letters and legends below Figures are self-explanatory. Uniform Figure legends: in Figures 2-5 there are symbols with rectangles, whereas in Figure 6 all the informations, i.e. name of bacteria and concentrations, are given below the plots. Uniform “Ref” and “B_longum10”. If the Authors choose the symbol rectangles, make them bigger, as it is hard to distinguish the 3 different B. uniformis concentrations.

R/ We decided to maintain the figure titles to offer a piece of introductory information about the figure content. However, we have shortened those titles to make them less redundant and tediously during reading. Additionally, we have increased the size of symbols included in respective plot legends for a better visualisation. Figure 6 was removed as it did not provide additional information with respect to that stated in the text.

-Figure 4 B-D. Concerning cytokine concentration in jejunum, what does it mean “pg/mL”? mL of what? Should instead the data be normalised to µg tissue, or proteins?

R/ The cytokine concentrations were expressed as pg/mL. A total of 100 mg of tissue was homogenised in 0.5 mL RIPA buffer, and a supernatant sample with an adjusted protein concentration (10 mg/mL) was used as final input in the assay. We have clarified this issue by in the text and figure legend of the new version (152-159 as well as in the legend of Figure 4).

-Figure 5. How exactly has been this ratio calculated? Should it be the ratio between the two cytokine concentrations? If so, it should be a dimensionless number, with no measure unit (pg/mL). Please, explain.

R/ As the reviewer mentioned, this is a simple ratio calculation; therefore, the Y-axis is dimensionless. We have corrected Figure 5 accordingly.
Results

3.2. Bacterial translocation
-Lines 239-241 and lines 396-398 of Discussion. The Authors should justify the cross-contamination. They cite in the Discussion only one reference as another example of cross-contamination, but it is just one study and a very very old one (year 1990).

R/ We have re-written those sentences in order to better express our thoughts (lines 242-244 and lines 396-399).

3.4. Cytokine concentrations in serum and illeum
-Line 278 and Figure 4A. Are the Authors sure that the difference between Control and B. uniformis 8 is not significant?

R/ We have reviewed the statistical analyses, and differences among groups were properly represented in the Figure 4 and Table 1.

-Line 306. Truncated sentence. Please, complete.

R/ The sentence was amended (line 292-293).

-Line 320. Is this result in accordance with other literature data?

R/ We have reviewed the scientific literature more deeply and found similar results. In general, few studies are showing TNF-alpha quantifications using in a similar manner as ours (method and samples), and those found at least did report serum values within the LOD of respective assays. We cannot discard that values of cytokine quantification in our study could be affected by multiple technical limitations attributed to low-sensitivity kit employed or to issues associated with the sample preparation, sample matrix, dilutions, buffers, etc., as previously reported when using Luminex based approaches (doi: 10.1016/j.vascn.2018.01.005, doi: 10.1016/j.jim.2013.04.009). Therefore, we have re-written lines 307-310 and 421-424 to give indications of the reasons why the TNF-alpha values could not be detected.

3.6. Effect of the intervention on the faecal microbiota composition
It seems a little bit strange that, except for Christensenellaceae and Bacillaceae families, the majority of data described in this paragraph are not shown in Figures/Tables. Why? In particular:

R/ We decided to present exclusively these data graphically because they were the only genera altered as described in the text and showing concordance with the findings at the family level. To include other microbiota features mentioned in the text, such OTUs, would make the graphical design more complex. Consequently, we have decided to omit the Figure 6 given the information presented in lines 344-355 clearly summarises the main results.

-Lines 328-321. No data shown on alpha diversity? Same comment applies to lines 344-358.

R/ Given that no meaningful differences were established among groups in any of the alpha diversity descriptors explored, we understand there is no need to show additional information and that it is sufficient to state it in the text, not even as a Supplementary Figure. This is a type of data representation presented in excess according to the reviewers’ thoughts. Moreover, the graphical representation of all the large variety of families, genera, and OTUs with differential abundance across groups is not feasible and would redundant to what already stated. All in all, we have decided to mention the taxonomic categories that showed a perturbation in the gut microbiota of the experimental groups based on a selected threshold for the log2-fold-change and adjusted (corrected) p-values and for those cases for which reliable identifications at least at genus level could be inferred (see lines 220-223 and lines 344-355).

Minor Revisions
Introduction
-Line 43. Delete the first “specific”, as it is repeated twice.

R/ Done as requested (line 49).

-Line 59. Change “with” to “and”.

R/ Changed as suggested (line 66).

-Line 71. Put superscript “109”

R/ Amended as requested (line 79).

Discussion
-Line 362. Change “its” to “their”.

R/ Corrected as suggested (line 360).

Reviewer 3 Report

Journal: Nutrients 
Manuscript ID: nutrients-680145 
Type of manuscript: Article 
Title: Safety assessment of Bacteroides uniformis CECT 7771, a symbiont of 
the infant’s gut microbiota  
Authors: Eva M. Gómez del Pulgar, Alfonso Benítez-Páez *, Yolanda Sanz 
Submitted to section: Prebiotics and Probiotics,

The present study performed an sub-chronic rats study for evaluating the usage of Bacteroides uniformis CECT 7771. They noted that no significant toxic effects were observed and does not raise safety concerns. 

I have some serious concern regarding this study.

First, based on the introduction research background: the Bacteroides uniformis CECT 7771 was originally isolated from faeces of healthy breastfed infants, which have been demonstrated to have a higher abundance of this Bacteroides species compared to formula-fed infants.

Please note that there are dramatical gaps between rodents and humans. I didn’t find any evidence that this bacteria strain can be found in rodents and I’m afraid the significance for the following human clinical trials.

I noticed significant decreases in the ALT level, although the increased ALT might be linked with the hepatic damages. I’m afraid that this change should be cautious, which might be associated with liver changes. Moreover, AST levels are missing.   

In figure 4, I noticed that IL-10 and IFN were decreased by Bacteroides uniformis administration. It is awkward to me, since their concentrations are pretty low, at pg/mL levels. Based on my experience, these concentrations are close to the bottom of the LOD.

Regarding the gut microbiota changes, my concern is authors omitted the gut microbiota compositions at the initial stage of this experiment. Moreover, authors also ignored to measure the B. uniformis abundance in the gut. If this bacteria was administration to the animals, the alpha diversity might be influenced by this single bacteria, and otherwise I’m afraid that this bacteria did not colonize properly in the gut.

Author Response

Reviewer#3 concerns:
The present study performed an sub-chronic rats study for evaluating the usage of Bacteroides uniformis CECT 7771. They noted that no significant toxic effects were observed and does not raise safety concerns.
I have some serious concern regarding this study.

First, based on the introduction research background: the Bacteroides uniformis CECT 7771 was originally isolated from faeces of healthy breastfed infants, which have been demonstrated to have a higher abundance of this Bacteroides species compared to formula-fed infants. Please note that there are dramatical gaps between rodents and humans. I didn’t find any evidence that this bacteria strain can be found in rodents and I’m afraid the significance for the following human clinical trials.

NOTE: Please be aware that line numbers should differ between the original MS-formatted version of the document submitted and the journal-formatted version.

R/ We thank the reviewer for her/his criticisms about our assessment. We are entirely aware of the gaps regarding the animal-to-human translational studies. We must say that the beneficial role and efficacy of B. uniformis CECT 7771 in obesity-associated dysfunctions have been extensively documented in previous pre-clinical studies. Before the evaluation of its efficacy in humans, it is recommended to accomplish the QPS characterisation according to guidelines defined by the European Food Safety Authority (https://www.efsa.europa.eu/en/topics/topic/qualified-presumption-safety-qps), the primary aim of this study. Consequently, we want to emphasise that beyond the potential health benefits unveiled here and in previous studies, the main goal of this assessment was to determine that oral consumption of this microorganism does not raise any safety concern. We have demonstrated its safety as food through a complete evaluation of different organ functions at physiology and molecular levels. Thus, this study represents a milestone to advance in the characterisation of B. uniformis CECT 7771 and to assess its probiotic potential clinically according to ISAPP indications. On the other hand, we have demonstrated that these animal species do harbour this microbial species (B. uniformis) in their guts. However, to provide further and independent proof of that, we have performed a PubMed search of rat studies where gut microbiota was assessed and found that several studies reported the presence of B. uniformis in rodents (e.g. doi: 10.1038/s41598-017-02880-0, doi: 10.1111/jam.12157, doi: 10.1111/bju.14553). Given the effects of B. uniformis CECT 7771 on obesity models in previous pre-clinical assessments, its safety profile proven in the present study, and the fact that this species is a symbiont in both rodents and humans, we consider that its future evaluation in humans holds promise. We also agree with the reviewer that direct extrapolation of effects from animals to humans cannot be done and that specific human trials will be needed to prove our hypothesis as indicated in the conclusions.

I noticed significant decreases in the ALT level, although the increased ALT might be linked with the hepatic damages. I’m afraid that this change should be cautious, which might be associated with liver changes. Moreover, AST levels are missing.

R/ We agree with the reviewer. ALT and ALP levels seemed to indicate no hepatic damage, and ALT levels could suggest a potential protective role of B. uniformis CECT 7771. Therefore, a more in-depth analysis would be required to determine to what extent the liver function is beneficially modified by oral administration of this bacterium and if it represents protection against liver damage. We have tone down our conclusions at this regard (405-410).

In figure 4, I noticed that IL-10 and IFN were decreased by Bacteroides uniformis administration. It is awkward to me, since their concentrations are pretty low, at pg/mL levels. Based on my experience, these concentrations are close to the bottom of the LOD.

R/ We have reviewed with the manufacturer manual the LOD for the cytokines quantified in a multiplex manner. The reviewer is right. The values are pretty close to the lower LOD. However, the cytokine quantification was assisted by a manufacturer proprietary software that produced accurate quantifications with no alerts of unreliable (out of LOD) measurements with passing all internal controls and calibrators properly. We found this type of kits more appropriate for measuring IL10, INF-gamma, and TNF-alpha concentration in serum because the sample amount limitations.

Regarding the gut microbiota changes, my concern is authors omitted the gut microbiota compositions at the initial stage of this experiment. Moreover, authors also ignored to measure the B. uniformis abundance in the gut. If this bacteria was administration to the animals, the alpha diversity might be influenced by this single bacteria, and otherwise I’m afraid that this bacteria did not colonize properly in the gut.

R/ We understand the reviewer's doubts concerning the above mentioned potential issues. Nevertheless, we must mention that we have applied SOPs previously published that have been adapted to microbiome research as regular routines demonstrating to minimise the inter-individual variability among individuals in animal models. These SOPs also improve the associations of specific microbiota signatures with treatments, thus attenuating the impact of confounding variables. In consequence, we expected to have no differences at baseline as previous reports indicate. We have re-phrasing the lines 106-110 for a better understanding of our experimental design. The results obtained when the alpha diversity parameters were explored do not necessarily support the absence of colonisation by B. uniformis CECT 7771. The treatment theoretically only would introduce one novel strain in the community (with minimal abundance when compared with the rest of the species). Furthermore, this species is expected to be already present in the ecosystem explored (see doi: 10.1038/s41598-017-02880-0, doi: 10.1111/jam.12157, doi: 10.1111/bju.14553). All in all, no drastic changes would be expected in terms of richness, evenness or dominance, as we retrieved. Those results contrast with that obtained from beta diversity evaluation (RDA included as requested by Editor – see lines 327-331) because both ecological approaches assess the microbial community at different levels.
Finally, we employed massive sequencing to explore global and particular changes in gut microbiota, including the administrated bacterial species, using respective and conventional microbial ecology descriptors. We did not quantify accurately B. uniformis CECT 7771 in the gut because the assessment was conducted primarily to provide proofs about its safety use during sustained oral administration. Additionally, we do not have samples available anymore to perform quantitative analyses (for example by qPCR) based on DNA and these will not be conclusively to determine the presence of living cells supporting colonisation. We hypothesised that the meaningful increase of OTUs associated with B. uniformis exclusively in the B_unif10 group and increasing of Bifidobacterium-associated OTUs in the B_longum10 group are quite likely the result of a transient colonization. Evidence supporting that notion can be reviewed in the new Supplementary figure 9 showing the multivariate exploratory assessment done (RDA), where samples from those groups are opposite distributed across the multidimensional space. Still, future experiments should be conducted to demonstrate these cells remain alive in the gut, thus demonstrating colonisation but this will need additional samples that are not available. Accordingly, we have added some statements to tone down our conclusions (lines 442-444).

Reviewer 4 Report

 Though there is previously published literature on the topic, current paper address safety issues which has been a major concern.

Author Response

Reviewer#4 concerns:
Though there is previously published literature on the topic, current paper address safety issues which has been a major concern.

R/ We firmly think that QPS status should be granted to a greater diversity of symbionts inhabiting the human gut. Therefore, our study is important to demonstrate the safety of microorganisms different from classical lactic-acid bacteria and bifidobacteria used so far.

Round 2

Reviewer 2 Report

All the raised concerns have been properly addressed by the Authors and the quality of the paper improved significantly.